# A Nut for Every Bolt: Subunit-Selective Inhibitors of the Immunoproteasome and Their Therapeutic Potential

**DOI:** 10.3390/cells10081929

**Published:** 2021-07-29

**Authors:** Eva M. Huber, Michael Groll

**Affiliations:** Chair of Biochemistry, Center for Protein Assemblies, Technical University of Munich, Ernst-Otto-Fischer-Str. 8, 85747 Garching, Germany

**Keywords:** immunoproteasome, inhibition, inflammation, autoimmune diseases, drug development

## Abstract

At the heart of the ubiquitin–proteasome system, the 20S proteasome core particle (CP) breaks down the majority of intracellular proteins tagged for destruction. Thereby, the CP controls many cellular processes including cell cycle progression and cell signalling. Inhibitors of the CP can suppress these essential biological pathways, resulting in cytotoxicity, an effect that is beneficial for the treatment of certain blood cancer patients. During the last decade, several preclinical studies demonstrated that selective inhibition of the immunoproteasome (iCP), one of several CP variants in mammals, suppresses autoimmune diseases without inducing toxic side effects. These promising findings led to the identification of natural and synthetic iCP inhibitors with distinct chemical structures, varying potency and subunit selectivity. This review presents the most prominent iCP inhibitors with respect to possible scientific and medicinal applications, and discloses recent trends towards pan-immunoproteasome reactive inhibitors that cumulated in phase II clinical trials of the lead compound KZR-616 for chronic inflammations.

## 1. Introduction

Protein homeostasis, i.e., balanced protein synthesis and degradation, is essential for cell division and a plethora of intracellular events, among them signal transduction pathways [1,2]. In this regard, protein breakdown has to be tightly controlled and timely regulated. In eukaryotes, proteins destined for degradation are tagged with polyubiquitin chains and targeted to the 26S proteasome, a sophisticated molecular machine [3]. Its 19S-cap recognizes and removes the ubiquitin moieties, and, by consuming ATP, unfolds and feeds the polypeptide chains into its proteolytic 20S core particle (CP) [4]. The latter subcomplex represents the key component of the ubiquitin–proteasome system and is the target of numerous drugs and natural products [5,6].

The CP is assembled of 14 α-type and 14 β-type subunits that are stacked in four seven-membered rings around a central pore, creating a barrel [7]. In eukaryotes, the heptameric rings are built of seven distinct α- and and seven distinct β-subunits following an α_1-7_β_1-7_β_1-7_α_1-7_ stoichiometry (Figure 1) [8]. The α-subunits form the entry gates to the catalytic chamber [9,10] and serve as docking sites for the β-subunits during proteasome assembly [11], as well as for the association with regulatory particles such as the 19S-cap [12,13]. Three out of the seven distinct eukaryotic β-type subunits assembled in a CP, namely β1, β2 and β5, are proteolytically active [14], and together with their adjacent subunits in the same β-ring form channels for polypeptide binding [8]. Because the subunits differ in their amino acid composition, these substrate channels vary in their physicochemical properties and give rise to distinct cleavage preferences. Being an endoprotease, each substrate binding channel is composed of primed and non-primed specificity (S) pockets that bind target polypeptides in the C- to N-terminal direction (Figure 2). The primed sites are rather shallow, maybe because they release the C-terminal cleavage product early during the reaction cycle. In contrast, the pronounced non-primed pockets allow for tight interactions with the N-terminal polypeptide segment and thereby largely determine cleavage specificity [8]. Between primed and non-primed sites, the scissile peptide bond is cleaved by the active site Thr1 residue (Figure 2), which classifies the proteasome as an N-terminal nucleophile hydrolase. The Thr1 residue is embedded in and activated by two essential hydrogen bond networks [15].

In archaea and simple eukaryotes, proteasome activity mainly serves for the production of oligopeptides that are further decomposed into single amino acids by the action of cytosolic peptidases and ultimately reused for *de novo* protein biosynthesis. In jawed vertebrates like mammals, a fraction of proteasomal cleavage products can escape further degradation by entering the endoplasmic reticulum (ER). After N-terminal trimming by ER-residing peptidases, the fragments associate with major histocompatibility class I (MHC-I) receptors [16]. The resulting complexes are transported to the cell surface and presented to patrolling immune cells [17]. If the immune system detects peptides of foreign, i.e., viral or bacterial, origin on a body cell, the immune defense will be activated and the antigen-presenting cell will be killed to prevent the spreading of the infectious agent [18,19]. This process demands for the efficient production and presentation of proteasomal cleavage products and for reliable discrimination of self- and non-self peptides by immune cells. To this end, mammals evolved specialized CP types that shape the adaptive immune system and take over tissue-specific functions in protein degradation [20,21,22].

Among the four mammalian CP types, the constitutive proteasome core particle (cCP) is the most abundant, as it is present in all body cells and performs the bulk of intracellular protein degradation. Its subunit composition and cleavage specificities are closely related to the proteasome of unicellular eukaryotes and it comprises the active entities β1c, β2c and β5c [23,24,25,26]. The second best characterized CP is the immunoproteasome. Although expression of the immunoproteasome core particle (iCP) can be induced in all body cells by the action of the pro-inflammatory cytokines interferon-γ (IFNγ) and tumor necrosis factor-α (TNFα), only immune cells constantly produce iCPs [20]. The iCP efficiently generates oligopeptides with hydrophobic C-terminal anchor residues and a high affinity for the MHC-I receptor [27,28], thereby supporting antigen presentation as well as pathogen clearance. This physiological function of the iCP is based on its altered substrate cleavage preferences emanating from the subunits β1i, β2i and β5i [29,30,31]. 

Only in recent years two additional CP types have been discovered; first, the thymoproteasome core particle (tCP), exclusively present in cortical epithelial cells of the thymus, drives maturation and positive selection of T cells by producing low affinity peptides for MHC-I receptors [32,33]. This process is crucial for establishing self-tolerance as well as preventing autoimmunity [34] and relies on the unique subunit composition of the tCP (β1i, β2i and β5t) [32]. The latest discovery of CPs was the spermatoproteasome (sCP), which is required for proper spermatogenesis in mammalian testes and incorporates a variant of subunit α4, thereby altering the interaction with regulatory particles [35,36,37]. Apart from these main CP types, a number of mixed particles have been described as well [38,39,40], however their physiological relevance remains to be investigated in more detail.

In the mid-90s of the last century, scientists started to explore CP inhibition [41]. At that time, it was unknown whether the crucial function of the CP for cell survival would leave a therapeutic window of application for CP inhibitors or not. To date, 30 years later, proteasome inhibition represents an incredible success story that has led to a plenitude of proteasome ligands, both from synthetic and natural sources, including three clinically applied inhibitors (for up-to-date reviews on proteasome inhibition see for example: [6,42,43]). Bortezomib (Velcade^®^), carfilzomib (Kyprolis^®^) and ixazomib (Ninlaro^®^) significantly improved the clinical outcome of multiple myeloma patients by non-selectively targeting CPs and killing cancer cells that strictly rely on high-capacity protein degradation [44,45]. The constant production of antibodies and the accumulation of misfolded immunoglobins sensitize multiple myeloma cells for proteasome inhibition and apoptosis [46]. However, the side effects associated with proteasome inhibition, as well as the high reactivity and cytotoxity of clinically applied compounds, limited additional as well as long-term therapeutic applications in non-malignant diseases so far [47,48]. During the last decade, with increasing knowledge about the iCP, selective inhibition of CP subtypes and single subunits was explored. Initial preclinical studies reported impressive results, suggesting that selective inhibition of the iCP could be beneficial for the treatment of autoimmune diseases, for which only few modestly effective therapeutic options are currently available. The intimate connection between the iCP and pathogenesis of inflammatory diseases or certain cancers opened up a new field of proteasome research and this review intends to cover the developments in this vivid area of academic and pharmaceutical research.

## 2. Principles of Proteasome Inhibition

### 2.1. Covalent Versus Non-Covalent Inhibitors

Inhibitors can be grouped into non-covalently and covalently acting ones depending on their type of interaction with the target protein. The former group acts ***per se*** reversibly, while covalent inhibitors can form permanent (irreversible) or transient (reversible) linkages with proteins. In general, non-covalent inhibitors are expected to cause less unwanted side effects and to have fewer off-targets. Moreover, they are rumored to be less toxic and target single subunits more selectively than their covalent counterparts, the selectivity of which may be obscured by off-target modifications after long-term exposure. Although non-covalently acting CP inhibitors have been investigated in detail (reviewed for example in [49,50]), none of them have been investigated in clinical trials so far. Most importantly, the high substrate turnover of CPs and the accumulation of substrates upon CP inhibition may interfere with tight and sustainable blockage, thus leading to early displacement of inhibitors from the active sites without inducing significant and therapeutically relevant effects [51]. Despite their high and often indiscriminate reactivity associated with off-target activities, covalent inhibitors are currently experiencing a revival in drug development [52] and appear to be the compounds of choice for CP inhibitors with clinical applications. Their advantages are high potency, lower effective doses, long-lasting effects by low dissociation rates or irreversible inhibition, favorable pharmacodynamic profiles (that endure detectable drug levels in body fluids), and less sensitivity to pharmacokinetics (e.g., resorption and metabolism) [53]. Although there are exceptions (see Section 4.1.1, Section 4.3.1 and Section 5.1), most proteasome inhibitors share a peptidic scaffold with a C-terminal electrophilic warhead that covalently targets the catalytic Thr1 residue (see also Section 2.2). Among the class of non-covalent CP inhibitors, non-peptidic ones are also known, which offer superiority in terms of metabolic stability and bioavailability. Of note, most natural and synthetic CP ligands target the non-primed substrate binding pockets that are more pronounced compared to their primed counterparts. In the following sections, inhibitors with all kinds of structures, functional groups and iCP subunit profiles will be presented and discussed in the context of their therapeutic potential.

### 2.2. Important Reactive Warheads

Various electrophilic head groups can covalently target the CP, including aldehydes, vinyl sulfones, vinyl amides, β-lactones, boronic acids and α, β-epoxyketones, among others. Their reaction mechanisms, as well as pros and cons, are discussed in detail elsewhere, for example in [5,49,54]. Here, only boronic acids and α, β-epoxyketones are introduced, as these are the electrophiles most frequently used for iCP inhibitor development. 

Boronates are a group of highly reactive, slowly reversible CP inhibitors. In this regard, a number of off-targets have been reported for bortezomib, the prototype boronic acid CP inhibitor and frontline treatment for multiple myeloma patients. In particular, co-inhibition of neuronal serine proteases by bortezomib has been correlated with severe neurodegeneration and neurotoxicity [55]. Although this off-target activity can be reduced as exemplified with ixazomib [56], oxidative deboronation and subsequent reaction with, for example, glutathione, might also cause unwanted effects [57,58].

With the discovery of the natural product epoxomicin and its electrophilic epoxyketone warhead, an alternative to aldehydes and boronic acids became available [59]. Modifying both the Thr1 hydroxyl group and its free N-terminus [60], the epoxyketone irreversibly inhibits proteasome activity without touching other intracellular serine or cysteine proteases [55]. This improvement in selectivity led to the approval of the second-generation CP inhibitor carfilzomib (Kyprolis®) for clinical use [61]. The epoxyketone warhead was also selected for the development of various peptide-based iCP selective inhibitors (see Section 4.3.2, Section 5.4, Section 5.5 and Section 5.6).

## 3. Inhibition of the Immunoproteasome—A Short Historic Outline

Along with the growing clinical success of the broad-spectrum proteasome inhibitor bortezomib (Velcade®) in the early 2000s [62] and the discovery of CP subtypes, the interest in selective CP inhibition rose. In 2009, the report of the iCP ligand ONX 0914 (formerly PR-957) and evidence for its therapeutic activity in mouse models of autoimmune disorders caused sensation [63]. The study provided the proof-of-concept for selective iCP inhibition and opened up new fields of application—in particular for diseases that are characterized by elevated iCP levels, such as chronic inflammatory syndromes, certain types of cancers and neurodegenerative diseases. Inspired by the hope for lower effective doses, less side effects and better safety profiles compared to conventional non-selective CP inhibitors, the development of and screening for subtype-selective CP ligands by academic groups and pharmaceutical companies soon picked up speed. In parallel, structural studies on the iCP were conducted to facilitate and support drug design [26,64,65,66]. This combined effort over the last decade led to the identification of numerous inhibitors for all active sites of cCP and iCP. Besides serving as excellent research tools to monitor proteasome activities [67] and to assess the contribution of single CP subunits to antigen processing, presentation [68] and cytokine production, these compounds allowed further exploration of the role of the iCP in disease progression and potential therapeutic applications [47]. Their respective chemical structures, their development, as well as their biological activities are thoroughly discussed in Section 4. However, as the therapeutic benefit of single-subunit-selective inhibition in preclinical models of autoimmune diseases turned out to be low, co-inhibition of at least two iCP subunits was aspired. In this regard, ONX 0914, initially considered as β5i-selective, has recently been re-classified as a pan-iCP inhibitor that targets both β5i- and β1i-subunits [69]. Section 5 will account for the current developments in the field of pan-immunoproteasome inhibition and highlight promising compounds that might find clinical applications in the future.

## 4. Subunit-Selective Inhibitors of the Immunoproteasome 

### 4.1. Inhibitors of Subunit β1i

Already in the 1990s it was noted that iCP and cCP differ in their substrate cleavage preferences. For iCPs, cleavages after acidic residues (caspase-like activity) were found to be reduced, whereas hydrolysis after hydrophobic (chymotrypsin-like (ChT-L) activity) and basic (trypsin-like activity) residues was enhanced [29,70]. This notion correlated with structure-based comparison of β1c- and β1i-sequences [8], indicating that key polar residues in the substrate binding channel of subunit β1c in cCPs are replaced by hydrophobic ones in β1i of iCPs. These findings allowed for the development of β1i-selective inhibitors long before structural data visualized the β1i-substrate binding channel for the first time [26].

#### 4.1.1. Non-Covalent Inhibitors

The only known non-covalent inhibitor with pronounced β1i-selectivity has been reported by Ettari et al.; amide compound 7 has a K_i_ value of 0.021 µM for β1i and retains a residual activity for other cCP and iCP subunits of > 17% at 50 µM (Table 1; Figure 3) [71]. Molecular docking indicates smooth fitting to the β1i/β2i-substrate binding channel similar to ONX 0914 [71], thus qualifying the compound as a suitable building block for future fragment-based drug design strategies.

#### 4.1.2. Covalent Inhibitors

In 2007, Millenium Pharmaceuticals Inc. filed a patent on peptide boronic acid inhibitors, among them a fluorinated β1i-selective derivative of bortezomib, termed ML604440 (Figure 3) [92]. Later, this dipeptide boronate was used to study differential antigen processing by the subunits β1c and β1i [72] and to evaluate therapeutic effects on immune thrombocytopenia, an autoimmune bleeding disorder. Peripheral blood mononuclear cells of patients suffering from immune thrombocytopenia showed increased expression of β1i, but treatment with ML604440 failed to normalize the number of platelets in mice with immune thrombocytopenia [93]. In contrast, ONX 0914 was able to revert platelet counts in this mouse model, suggesting that selective inhibition of β1i may not be sufficient to reach therapeutic efficacy [93].

In 2009, UK-101, a structural derivative of dihydroeponemycin, was reported to be a β1i-selective inhibitor [74]. Its aliphatic *n*-heptanoic tail is directed to the interface of β1i- and β2i-subunits and the tert-butyldimethylsilyl group hinders modification of β5-subunits (Figure 3) [94,95]. UK-101 targets subunit β1i in Raji cell lysates with an IC_50_ value of 0.104 µM, and shows 144-fold selectivity over β1c (Table 1), and 10- and 30-fold selectivity over β5c as well as β5i, respectively. This modest subunit selectivity profile resembles that of the parent molecule dihydroeponemycin [94,96]. UK-101 has been shown to inhibit proliferation of prostate cancer cells [74,97] and tumor growth in a xenograft mouse model of prostate cancer [97], but this effect may result from partial co-inhibition of β5i- and β5c-active sites at the applied concentrations [73,75]. In agreement, off-target activities towards other proteasome subunits have been observed over time and these may result from cleavage of the ether bond by an intracellular hydrolase and loss of the tert-butyldimethylsilyl group, resulting in progressive co-inhibition of β5-active sites [98].

Based on NC-001, a potent β1-inhibitor [99], the Overkleeft group developed a β1i-selective epoxyketone. The tetrapeptide LU-001i features a fluorinated l-Pro analogue at P3 (Figure 3) and has similar potency to UK-101 but higher subunit selectivity. IC_50_ values were determined in Raji cell lysates as 0.095 µM for β1i and 24 µM for β1c (252-fold selectivity for β1i over β1c), and ≥20 µM for all other proteasome subunits (Table 1) [73]. Being active in cells, the therapeutic potential of LU-001i was evaluated in vitro and in vivo. These studies revealed that LU-001i alone is unable to block differentiation of T helper cells or lower MHC-I surface and cytokine expression [91]. Most importantly however, it lacks the potential to ameliorate symptoms of dextran sulfate sodium-induced colitis in mice [91].

Recently, another epoxyketone, dubbed DB-310, has been reported to be a potent and selective β1i-inhibitor (Figure 3). It is claimed to have improved selectivity and permeability in cells that overexpress the efflux transporter ABCB1—a requirement for brain barrier penetration. With its moderate β1i-selectivity (~8-fold), DB-310 suppresses proinflammatory cytokines and leads to cognitive improvements in a mouse model of Alzheimer’s disease [75].

The most potent and selective β1i-inhibitor known to date, KZR-504, has been described by Kezar Life Sciences Inc. (Figure 3). IC_50_ values were determined in MOLT-4 (human T cell leukemia) cell lysates as 0.051 µM for β1i and 46.35 µM for β1c (908-fold selectivity for β1i over β1c; Table 1) [76]. Although KZR-504 has been improved with respect to solubility and stability, it failed to suppress the production of pro-inflammatory cytokines in in vitro cell culture assays [76], similar to LU-001i [91].

Given the shallow S2-pockets of proteasome substrate binding channels [26], the presented β1i-selective ligands are diverse in their P2-sites but share some structural features at their P1- and P3-sites. First, Leu (UK-101, DB-310 and ML604440), cyclohexyl (LU-001i) and Phe (KZR-504) moieties are reasonable P1-sites to target β1i. Second, most β1i-selective compounds share a cyclic residue at P3—a structural feature that is also known from the peptide substrate Ac-PAL-AMC, used to selectively monitor β1i-activity by fluorescence spectroscopy [100]. Like Ac-PAL-AMC, LU-001i and DB-310 carry a P3-l-Pro residue, while KZR-504 and ML604440 feature a 2-pyridone or a 2-trifluoromethylbenzoyl moiety, respectively. Although cyclic residues appeared to drive β1-selectivity in general [73,76], the reason for this observation was initially unclear. In 2015, a structural study showed that compounds with Pro at P3 are tilted in β2/β3- and β5/β6-substrate binding channels compared with their Leu analogues. However, no such effect has been observed at the β1/β2-active sites. Particularly, Asp114 of β3 and β6 as part of the β2/β3- and β5/β6-substrate binding channels displaces P3-Pro ligands and leads to their pronounced β1-selectivity [101]. Similar findings were recently reported by Johnson et al., who showed by molecular modelling that the 2-pyridone moiety of KZR-504 fails to interact with Asp114 of β3- and β6-subunits [76].

### 4.2. Covalent Inhibitors of Subunit β2i

Historically, subunit β2c of cCPs has been assigned trypsin-like activity, but in fact it is rather promiscuous, because its S1-pocket, the major determinant of substrate specificity, is lined with Gly45 and thus is rather spacious [8]. Subunit β2i of iCPs is highly similar to its constitutive counterpart β2c in terms of structure and activity [26], raising questions about the biological function and benefit of β2i versus β2c. Additionally, the structural similarity precludes development of subunit-selective fluorogenic substrates [100] and inhibitors. 

Nonetheless, after huge screening efforts, some progress in this field was recently reported. Synthesis and evaluation of a library of ONX 0914 derivatives identified the β2i-selective compound LU-002i (Figure 3) [77]. ONX 0914 served as a blueprint, as it slightly discriminates between β2i and β2c (~1.8 fold preference for β2i) according to its subunit selectivity profile [73]. By installing large P1-sites on the otherwise unchanged ONX 0914 scaffold, LU-002i with pronounced β2i-selectivity and potency was obtained (IC_50_ values determined in Raji cell lysates: 0.18 µM for β2i; 12.1 µM for β2c; β2c/β2i ratio: 67; Table 1). To visualize the structural basis for this selectivity of LU-002i, yeast proteasomes having incorporated large parts of human β2c- or β2i-subunits were designed and produced [77]. Complex structures with a derivative of LU-002i revealed that the β2i-active site can adapt more easily to bulky P1-residues, whereas the corresponding site in β2c seems to be rather rigid. In particular, the hydrogen bond interaction between Asp53 and His35 in β2c might oppose the binding of inhibitors staffed with large P1-sites, while Glu53 of β2i might allow for more flexibility and leave a narrow window to establish selectivity [77]. As an add-on, LU-002i has been functionalized at the P2-site to create an activity-based probe (ABP) for the visualization of β2i-activity of human iCPs. Although LU-002i and its ABP derivative are functional in cell lysates, it is unknown whether they can penetrate cell membranes and are effective in vivo. To the best of our knowledge, LU-002i is the only β2i-selective ligand known to date and it has not yet been tested for any biologically or therapeutically relevant in vivo effects.

### 4.3. Inhibitors of Subunit β5i

The ChT-L activities of β5-subunits are by far the most important cleavage specificities of CPs and thus they are the target of most synthetic and natural proteasome inhibitors. As with β2-subunits, the β5c- and β5i-substrate binding channels only marginally differ in sequence and structure. The most pronounced distinction is the size of their S1-specificity pockets, originating from a conformational change of Met45. While in subunit β5c Met45 flexibly arranges with various kinds of apolar amino acid side chains, preferentially leucine, the S1-pocket of β5i is enlarged and promotes cleavage after bulky hydrophobic residues such as Phe, Tyr or Trp [26]. Thus, by staffing inhibitors with large P1-residues, β5i-selectivity can be gained. 

Given the huge number and the structural heterogeneity of β5i-selective drugs published over the last years, their presentation and discussion is grouped according to mode of inhibition and structural features.

#### 4.3.1. Non-Covalent Inhibitors

In the hope for higher isoform selectivity compared to covalently acting compounds, several non-covalent and often non-peptidic iCP inhibitors with exquisite potency and β5i-selectivity have been identified over the last few years. While some already underwent several optimization steps, the smaller ones especially might serve as potent fragments for future extensions and modifications.

##### Isolated Fragments

In a computational docking approach with ~300,000 compounds, a β5i-selective non-peptide fragment (compound 1; Figure 3) was discovered. Using purified human iCP, the IC_50_ value for β5i was determined as 1.7 µM. At 5 µM, β5c showed 80% residual activity, indicating that the compound preferentially targets β5i over β5c (Table 1) [78]. However, other proteasome activities have not been tested for co-inhibition, thus leaving questions about the selectivity. Further evaluation, e.g., by structural studies, is necessary to confirm the inhibition and to make use of the findings in rational drug design efforts.

##### Psoralenes

Another virtual screening identified psoralenes as potential iCP inhibitors. The nonpeptidic compound 3 (Figure 3) targets β5i with 107-fold selectivity over β5c (K_i_ values of 1.6 µM for β5i and 172.2 µM for β5c; Table 1) [79]. It penetrates cell membranes and inhibits the iCP in cells with similar selectivity as ONX 0914 but 1000-fold lower potency. To enhance target affinity, compound 3 was functionalized with different electrophilic warheads [79] and further modifications were introduced to stimulate selectivity [102]. To make use of the psoralene scaffold in future fragment evolution or linking approaches, structural analysis of the binding mode of compound 3 or one of its derivatives to the CP might be useful.

##### N,C-Capped Dipeptides

In recent years, several non-covalently acting N,C-capped dipeptide and dipeptidomimetic inhibitors with β5i-selectivity have been reported [100,103]. Among them, the N,C-capped dipeptide DPLG3 (Figure 3). It targets subunit β5i with an IC_50_ value of 0.0045 µM and 7200-fold selectivity over β5c, while other activities are inhibited <50% at 33.3 µM (Table 1) [80]. DPLG3 suppresses cytokine release in vitro and shows efficacy in an animal heart transplantation model by significantly extending the graft survival time [80]. Although DPLG3 should not target subunits other than β5i at the applied concentrations, in vivo inhibition values have not been reported and so it remains to be further investigated whether single-subunit inhibition is really sufficient for the observed therapeutic effects [47]. Due to its adverse physicochemical properties, i.e., high hydrophobicity and poor water solubility, as well as poor cell penetration ability, efforts to improve DPLG3 were undertaken and afforded several derivatives, among them PKS21272 with 500-fold β5i-selectivity and IC_50_ values of 0.0012 µM for β5i and 0.61 µM for β5c, as well as < 50% inhibition of other subunits at 33.3 µM [104]. This compound is cell-penetrating and inhibits proteasomes in vivo but at lower selectivity (81-fold selectivity for iCP) and potency (0.053 µM). In addition, PKS21272 can inhibit activation and proliferation of T cells [104]. Notably, derivatives of DPLG3 are claimed to be ‘active site-directed non-competitive inhibitors’ [64,104], meaning that their inhibition potency is not affected by accumulating substrate concentrations. Considering their peptidic or peptide-like structure and their binding to the natural β5-substrate binding pockets [64], a non-competitive mode of action is questionable.

##### Thiazole Compounds

In 2014, Roche Diagnostics patented a set of non-peptidic compounds with pronounced β5i-selectivity and -potency [81]. This set of inhibitors is based on a thiazole core that features an aromatic bicyclic moiety at the C2-site linked to a carboxylic acid via a peptide bond. To resolve the binding mode of the ligands to the proteasomal active site and their basis for selectivity, the most potent (IC_50_ value of 0.025 µM for human β5i) and selective (800-fold over β5i) compound of this series [81], termed Ro19 (Figure 3), was studied by X-ray crystallography with a humanized yeast proteasome, featuring the human β5i/β6-substrate binding channel [105]. The X-ray data visualized that Ro19 occupies the S1-pocket and also S1/S3-sub-sites that are not accessible to natural peptide-based ligands but various synthetic compounds [106,107]. Ro19 and all other compounds of this class are β5i-selective, as their bulky quinolone moiety complies better with the spacious S1-pocket of β5i than with the smaller one of β5c. In addition, binding to the active sites of β1i and β2i is precluded by steric hindrance of the thiazole ring with residues of their S3-pockets. Notably, Ro19 is a poor inhibitor of mouse β5i—probably because the mouse specific residue Met31 hinders its placing [105]. Proteasomal substrate binding channels are well conserved in sequence and structure among different species, but sub-pockets outside the natural ligand binding sites may not be. Thus, inhibitors with oversized P1-sites or non-standard binding modes like Ro19 may be associated with significant species-selectivity and demand for careful selection of animal models for preclinical studies. Similar observations were made with the covalent β5i-inhibitors PR-924 [106] and M3258 [66] (see Section 4.3.2.)

#### 4.3.2. Covalent Inhibitors

The majority of β5i-inhibitors feature a reactive functional group that either reversibly or irreversibly links them to the nucleophilic Thr1 residue (see Section 2.2). However, this covalent mode of action usually gives rise to co-inhibition of unwanted activities and often dampens subunit selectivity, because all proteolytically active β-subunits of the CP share the same active site architecture and catalytic mechanism [8,15,26]. Alternative approaches therefore explore non-catalytic amino acids nearby the active site as nucleophiles.

##### Inhibitors targeting Cys48 of β5i

The concept of targeting non-catalytic residues has been successfully applied to a wide range of enzymes, including the proteasome [108,109]. More recently, it has also been exploited for the development of β5i-selective inhibitors. Three independent studies selected Cys48 at the interface of the S2- and S4-specificity pockets as a nucleophilic target residue [82,83,110]. Because β5c encodes a non-reactive Gly at this position [26], the strategies aimed for a strong covalent blockage of β5i and either no or a modest non-covalent inhibition of β5c. The concept of gaining subunit preferences by reactivity rather than non-covalent interactions is a smart approach pursued by several pharmaceutical companies.

Already, in 2015, Principia Biopharma Inc. filed a patent describing inhibitors that form a reversible covalent bond with the conserved non-catalytic Cys48 of β5i [110]. The prototype compound PRN1126 (Figure 3) was shown to be ~30-fold selective for human β5i over human β5c (IC_50_ values determined for purified human cCP and iCP: 0.0072 µM for β5i and 0.21 µM for β5c) without touching other proteasome activities (Table 1) [69]. Although being truly β5i-selective, PRN1126 dampened neither MHC-I surface expression nor interleukin (IL)-6 secretion, nor Th17 differentiation. In addition, no therapeutic benefit was noted in experimental models of autoimmune diseases [69], questioning the hitherto existing dogma that selective inhibition of β5i translates into therapeutic potential. Notably, experimental proof for the proposed reaction mechanism of PRN1126 with Cys48 is still not available, but the nitrile functional group of PRN1126 (Figure 3) is supposed to approach Cys48 via the S2-pocket. Subsequent nucleophilic attack of the C-N triple bond by the thiol likely results in a reversible covalent thioimidate linkage [111].

Furthermore, a patent application of Merck KGaA from 2016 reports a group of β5i-selective compounds with IC_50_ values <0.050 µM and ≥150-fold selectivity for β5i over β5c [82]. The compounds bear a boronic acid head group that targets Thr1, and a second electrophile—frequently an acrylamide moiety (such as in compound 16, Table 1, Figure 3)—that may function as a Michael acceptor in the reaction with Cys48 of subunit β5i. Given their dipeptide structure, these boronic acid inhibitors are expected to target Cys48 like PRN1126 via the S2-pocket, but experimental proof for this reaction mechanism is not available.

In another study, Dubiella et al. designed a decarboxylated peptide that is derived from carfilzomib (Kyprolis®) and directed to Cys48 via an electrophile at its P4-site [83]. Among the tested nucleophiles, the α-chloroacetamide variant 1-CA was the most potent. Variant 1-CA showed nine-fold selectivity for β5i over β5c, and did not target other subunits of human cCP or iCP [83]. Using a mutant yeast proteasome, the covalent reaction of the α-chloroacetamide at P4 with Cys48 at the active site of β5 was visualized and confirmed. Notably, yeast β1- and β2-subunits also encode potential nucleophiles in their substrate binding channel, but in agreement with activity assays neither Ser48 of β1 nor Thr48/Cys31 of β2, nor Cys114 of β3 reacted with 1-CA [83]. This notion is also in agreement with the subunit selectivity of PRN1126 [69]. Improvements of the peptide moiety of 1-CA finally led to 4-CA (Figure 3). Featuring a Asn residue at P3 that can hydrogen bond to Ser27 of β5i but not to Ala27 of β5c, 4-CA showed improved β5i-potency (IC_50_ 0.64 µM) and -selectivity (156-fold over β5c; Table 1) [83].

##### Inhibitors Targeting Thr1

Oxathiazolones

Oxathiazolones are assumed to irreversibly modify the proteasomal active site. Multiple compounds of this class have been investigated for their β5i-selectivity, with HT2004 performing the best with purified human CPs (k_inact_/K_i_ for β5i: 1093; selectivity for β5i over β5c: 4752; Table 1, Figure 3) [48]. Although oxathiazolones are able to penetrate cell membranes, their instability in water prevented further biological studies [48]. The oxathiazolone head group has also been installed on non-covalent psoralene fragments, ultimately resulting in a potent and selective inhibitor (IC_50_ for purified human CPs: 0.106 µM for β5i; residual activity of other active sites is ≥ 67% at 10 µM) [79,102]. Despite significant improvements, poor cell penetration prevented further investigations [102].

Piperlongumine

Piperlongumine is a pepper alkaloid that features a vinylamide group similar to CP inhibitors of the syrbactin class (Figure 3). By structural analogy, piperlongumine was proposed to irreversibly modify the active site Thr1 of proteasome subunits in a Michael-type 1,4-addition [84,112]. In enzyme assays with human cCP and iCP, piperlongumine was found to block β5i with an IC_50_ value of 15 µM, while β5c and β1c were inhibited <30% at 50 µM (Table 1). However, other catalytic sites of the proteasome were not tested for potential co-inhibition. The cytotoxic effects of piperlongumine have been linked to various molecular mechanisms, including oxidative stress, DNA damage and apoptosis [113], suggesting that iCP inhibition might only in part contribute to this biological activity [84].

Boronates

In the search for boronic acids that are not bioactivated by oxidative deboronation [58] and carry a non-peptidic scaffold, a virtual compound library was screened by molecular docking. Hits that were predicted to adopt a Ro19-like binding mode [105] were prioritized. Among the identified compounds, 1a (Figure 3; originally published as 1) is ranked the highest, as it shows modest affinity for β5i (IC_50_ value for purified human iCP: 34 µM) and moderate selectivity over β5c (three-fold; Table 1) [85]. However, a complete cCP and iCP subunit selectivity profile has not been determined. Considering the high reactivity of boronic acid compounds in general and the low potency of 1a, the potential of this agent as a CP ligand is considered rather low, but the scaffold might inspire future fragment-based inhibitor design strategies.

In a joint venture, Genentech Inc. and Proteros Biostructures GmbH determined the X-ray structure of the human iCP in complex with a β1i- and β5i-selective boronic acid [65] (see also Section 5.3), patented by Roche [90]. Based on these structural data, the highly β5i-selective compound 22 (IC_50_ value for purified iCP: 0.0041 µM for β5i; 2219-fold selective over β5c; Table 1, Figure 3) was designed. Yet, β2-inhibition values have not been reported. The inhibitor did not induce cell death in immune cells, but was not tested for its potential to lower proinflammatory cytokines [65].

Besides, peptidic boronates with β5i-selectivity have been reported by Merck KGaA. In addition to the bifunctional ones discussed above [82], the company developed M3258 (Figure 3), an orally-available β5i-selective inhibitor with 614-fold selectivity over β5c and all other catalytic sites of cCP and iCP (IC_50_ value for purified iCP: 0.0041 µM) and favorable pharmacokinetic properties [66,114]. M3258 carries a bulky 3-benzofuranyl moiety that, together with the single amide group, dictates β5i-selectivity [115]. M3258 has strong anti-tumor activity in several multiple myeloma models, including those refractory to bortezomib, and reduces various cytokines, including IL-6 and TNFα levels [66]. Remarkably, the agent has fewer side effects compared with its structural analogues bortezomib (Velcade^®^) and ixazomib (Ninlaro^®^) that target both β5c- and β5i-subunits [66,116]. It is currently under evaluation in a phase I study as a single agent or combination therapy with dexamethasone for the treatment of patients with relapsed refractory multiple myeloma (NCT04075721) [66].

Peptide epoxyketone inhibitors

In 2009, scientists of Proteolix Inc. published the β5i-selective compound PR-924 [86]. PR-924, a tripeptide epoxyketone inhibitor carrying a non-natural d-Ala residue at P3 (Figure 3), has high potency and selectivity for β5i (IC_50_ values: 0.022 µM for β5i, 2.9 µM for β5c, 8.2 µM for β1i and >30 µM for β1c, β2c and β2i; β5c/β5i ratio: 131) [86], though it was not tested for effectiveness in autoimmune diseases. Instead, it was reported that PR-924 selectively induces cell death of human multiple myeloma cell lines and suppresses grafted multiple myeloma in mice [117]. However, these anti-proliferative effects required high drug concentrations that were likely not compatible with isoform-selective CP inhibition [118].

Despite these unsatisfactory results, the unusual stereochemistry of PR-924 and its exceptional β5i-selectivity were further investigated by academic groups. Derivatives of PR-924, combining various P1- and P2-residues as well as N-cap structures with a d-Ala residue at P3, were designed and evaluated for their subunit selectivity. Among them, LU-035i emerged as the most potent and selective compound (IC_50_ values in Raji cell lysates: 0.011 µM for β5i; β5i-selectivity: 500 over β5c; Table 1) [73]. In the course of this study it was noted that inhibitors with a P3-d-Ala residue and a 3-methyl-1*H*-indene N-cap, like PR-924 and LU-035i, display high β5i-selectivity (Table 1). Notably, this effect was not based on enhanced β5i-potency, but rather due to poor inhibition of β5c. As initial structural data could not explain this difference in performance [73], X-ray crystallographic data with humanized yeast proteasomes were collected. The structures visualized distinct binding modes for PR-924 in β5c/β6- and β5i/β6-substrate binding channels, respectively [106]. Consistent with the non-natural stereochemistry at P3, PR-924 adopts a kinked conformation in humanized β5i-proteasome structures, with the N-cap occupying a S3*-sub-pocket that cannot be approached by natural ligands. By contrast, at the β5c/β6-active site and in yeast CP crystal structures, PR-924 adopts a linear conformation that is assumed to be energetically disfavored and associated with poor affinity, thereby causing enhanced β5i-selectivity [106]. This effect applies to all derivatives of PR-924 with a P3-d-Ala residue and a 3-methyl-1*H*-indene cap, and was also found by two independent modelling studies [87,119]. Inhibitors with smaller N-caps bind to β5c and β5i in their kinked mode and thus lose β5i-selectivity. This result also indicates that the S3*-sub-pocket is larger in β5i than in β5c, although no pronounced structural differences could be spotted so far to explain the distinct binding modes. Furthermore, PR-924 and its derivatives show high selectivity for human β5i (>130-fold over β5c) but significantly lower preference for mouse β5i (16-fold over mouse β5c). While Val31 of human β5i is compatible with PR-924 binding, Met31 of mouse β5i sterically interferes with the kinked binding mode of PR-924, as well as its derivatives [106]. Usually, most inhibitors target proteasomes from distinct species with comparable potency, as the substrate binding channels are similar in primary, secondary and tertiary structure. However, when targeting non-canonical sites aside from the natural substrate binding pockets, the amino acids are not that well conserved. This fact complicates drug development and evaluation in preclinical studies, as different species can produce different outcomes. Therefore, animals for preclinical trials have to be carefully chosen and any disagreement in the amino acid lining of inhibitor target sites needs to be evaluated beforehand. In this regard, the ligand binding site of the target protein should be studied by structural means in early drug development stages.

Only recently, Kezar Life Sciences Inc. reported another β5i-selective inhibitor (compound 8 (Figure 3, Table 1); IC_50_ values determined in MOLT-4 cell lysates: 0.034 µM for β5i; 78-fold selectivity for β5i over β5c) [87]. Notably, compound 8 failed to control cytokine release in in vitro studies and disease progression in mice suffering from active collagen antibody-induced arthritis [87]. This report further substantiates the suspicion that single-subunit-selective inhibitors are poorly active as therapeutic agents (see also Section 5).

### 4.4. Therapeutic Potential of Subunit-Selective iCP Inhibitors

The identification of numerous highly subunit-selective inhibitors and more sophisticated methods to assay proteasome activities in vitro and in vivo allowed for a more thorough analysis of the biological effects and medicinal potential of iCP inhibition. These studies questioned the initial assumption that blockage of a single iCP subunit like β5i is sufficient to cause therapeutic benefits in autoimmune diseases.

Among the β1i-inhibitors, ML604440, KZR-504 and LU-001i failed to generate therapeutically relevant responses [76,87,91,93], and the anti-cancer activity reported for UK-101 [97] likely arises from co-inhibition of other CP activities [98]. Whether the beneficial effects of DB-310 in Alzheimer’s disease result from co-inhibition of other proteasome subunits as well, or from disease-specific circumstances, needs to be further explored [75]. In this regard, it is worth mentioning that most compounds are evaluated for their efficacy in different disease models and distinct settings, precluding any systematic comparison and reliable judgement of their bioactivity. For example, in the central nervous system, iCP inhibition has recently been correlated with disease worsening instead of improvement, thus demanding for careful selection of preclinical models [120].

Regarding β5i-selective inhibitors, PRN1126, compound 8 (Kezar Life Sciences Inc.) and compound 22 (Genentech Inc.) were found to be therapeutically inactive [65,69,87]. Whereas the reason for failure of PRN1126 and compound 8 was attributed to their single-subunit inhibition profile [69,87], Genentech Inc. argued that immune cells undermine iCP inhibition by inducing cCP expression, and thus escape apoptosis [65]. Notably, the study of Genentech Inc. aimed at inducing cell death in immune cells, while others including Kezar Life Sciences Inc. looked for a reduction in cytokine production without compromising cell viability. Although the outlined discrepancies may result from differences in read-out, target applications, applied concentrations etc., further evaluations are certainly required. For PR-924 [117] and M3258 [66], pronounced anti-cancer activity has been described in multiple myeloma tumor models. However, the effects observed for PR-924 are suspected to originate from co-inhibition of other CP activities [118]. It thus remains to be evaluated whether the activity of M3258 indeed results from single-subunit inhibition or from a high β5i-dependence of multiple myeloma cells as proposed by the authors [66]. In summary, with few exceptions, iCP inhibitors with high single-subunit selectivity appear to not have the expected therapeutic effects in vitro and in preclinical animal models of autoimmune diseases [47].

## 5. Pan-Immunoproteasome Reactive Inhibitors

Increasing evidence for the therapeutic failure of single-subunit-selective iCP inhibitors in inflammatory diseases promoted the search for compounds that target at least two iCP subunits. Specifically, co-inhibition of β1i and β5i, or β2i and β5i, appears to drive anti-inflammatory activity [87]. Based on these findings, the collection of known pan-reactive iCP inhibitors was recently supplemented by new compounds and will certainly grow further in the future.

### 5.1. Non-Covalent Amides

Ettari et al. recently published the non-covalent amide 6 (Figure 3) that targets β1i- and β5i-subunits of the iCP (K*_i_* values for purified human CPs: 4.9 µM for β1i and 4.4 µM for β5i; Table 1) and inhibits proliferation of a dexamethasone-resistant human multiple myeloma cell line (EC_50_: 17.8 µM) [88]. As inhibition values for the corresponding cCP subunits have not been determined, the iCP selectivity of amide 6 is elusive and remains to be proven.

### 5.2. IPSI-001

Initially declared as a β1i-specific inhibitor, IPSI-001 was one of the first iCP-selective compounds. This peptide aldehyde, also known as calpeptin (Figure 3), was shown to target subunit β1i of the iCP and induce apoptosis in multiple myeloma cells [89], but later this bioactivity was claimed to result from co-inhibition of β5c- and β5i-subunits [86]. Detailed and accurate evaluation of the subunit selectivity profile of IPSI-001 is essential to assess its potential as pan-reactive iCP inhibitor. To this end, further optimizations with respect to potency, stability and cross-reactivity might be required. So far, the low potency of IPSI-001 precluded in vivo studies [89]. Moreover, its aldehyde head group is prone to oxidative inactivation and well known to co-inhibit serine and cysteine proteases [121]. The latter feature can be beneficial to potential applications due to synergistic effects, or detrimental because of unwanted side effects.

### 5.3. Boronates

A patent filed by Roche reports about substituted triazole boronic acid compounds, among which ligand 1b (Figure 3; originally published as 1) has pronounced β1i- and β5i-selectivity (IC_50_ values of 0.002 µM for β5i and 0.039 µM for β5c, as well as 0.004 µM for β1i and 0.32 µM for β1c) [90]. Based on this report, Genentech Inc. and Proteros Biostructures GmbH created derivatives of 1b that are even more selective β5i-/β1i-inhibitors [65]. The compounds were not cytotoxic to immune cells, but anti-inflammatory activity was not examined.

### 5.4. LU-005i

A screen of ONX 0914 derivatives for improved β5i-selectivity led to the identification of LU-005i. Instead of a Phe residue, LU-005i features a cyclohexyl moiety at P1 (Figure 3) [73]. It shows activity towards subunits β1i, β2i and β5i with decent selectivity over the respective constitutive subunits (IC_50_ values for purified CPs: 0.052 µM for β1i, 0.47 µM for β2i and 0.16 µM for β5i; β1c/β1i ratio: >19; β2c/β2i ratio: 6; β5c/β5i ratio: 18; Table 1) and thus is considered a pan-immunoproteasome-selective inhibitor [91]. As the first reported compound that targets all three iCP subunits over the respective constitutive subunits, LU-005i was probed for its biological activity and therapeutic potential. These studies revealed that LU-005i is able to lower MHC-I surface expression and cytokine production. Moreover, LU-005i blocks differentiation of Th17 helper cells and alleviates symptoms of dextran sulfate sodium-induced colitis in mice [91].

### 5.5. ONX 0914

Reported in 2009, ONX 0914 (formerly PR-957) was the first CP inhibitor with moderate β5i-selectivity (nine-fold over β5c) and emerged as the lead compound for iCP inhibitor development in general (Table 1, Figure 3) [63]. In vitro, it reduces MHC-I expression, presentation of β5i-dependent antigens, as well as cytokine production and blocks Th17 differentiation [63]. ONX 0914 selectively induces apoptosis of CD14^+^ monocytes, thereby reducing IL-23 production and suppressing Th17 differentiation [122]. The potential of ONX 0914 to serve as a new treatment of chronic inflammations and autoimmune diseases was initially shown in mouse models of rheumatoid arthritis and diabetes mellitus type I [63], and later confirmed in a plethora of other preclinical studies, including experimental colitis [123], myasthenia gravis [124], Hashimoto’s thyroiditis [125], systemic lupus erythematosus [126], immune thrombocytopenia [93], experimental neuritis [127], autoimmune myocarditis [128] and multiple sclerosis [120,129] (for a complete list see [47]). In addition, as exemplified with ONX 0914, the iCP might qualify as a target to prevent transplant rejection [130], graft-versus-host disease [131] and pathogen-induced immunopathology [132]. Despite this excellent performance without cytotoxic side effects [133], unfavorable solubility properties of ONX 0914 prevented its evaluation in clinical trials [87].

The beneficial effects found with ONX 0914 were initially attributed to its β5i-selectivity. The development of inhibitors with improved subunit selectivity profiles, however, revealed that β5i-selective compounds are inferior to ONX 0914 with respect to biological activity in autoimmune diseases. Reassessment of ONX 0914 finally uncovered a substantial co-inhibition of ~60% of subunit β1i and marginal β2i-blockage at efficacious doses [87]. Most likely, this combined iCP subunit inhibition, or at least partial blockage of a second site, is required for the anti-inflammatory features of ONX 0914 [87]. Experimental proof for this theory was provided by assays with combinations of subunit-selective inhibitors. ML604440/LU-001i (β1i) and PRN1126 (β5i) [69], as well as KZR-504 (β1i) and compound 8 (β5i) [87], efficiently suppressed cytokine expression, while the single agents or combinations of β1i- and dual β2c/β2i-inhibitors failed. In fact, a combination of compound 8, KZR-504 and a β2-inhibitor, mimicking pan-iCP reactivity, was most effective but also killed 30% of stimulated immune cells [87]. While in the beginning ONX 0914 was the only available compound to study physiological effects of iCP inhibition, a plenitude of compounds is now available to assess all kinds of subunit combinations in detail for their bioactivity and to balance the right inhibition strength for the desired application. This is particularly important, as ONX 0914 was found to have adverse effects on disease progression when applied in the chronic phase of a mouse model of multiple sclerosis [120].

### 5.6. KZR-616

In an effort to find a compound with a similar subunit selectivity profile as ONX 0914 but improved physico-chemical properties for clinical applications, Kezar Life Sciences Inc. (formerly Onyx Pharmaceuticals Inc.) developed KZR-616 [87]. KZR-616 potently co-inhibits subunits β1i and β5i of iCPs (IC_50_ values for human cell lysates: 0.039 µM for β5i and 0.131 µM for β1i; β1c/β1i ratio: >80; β5c/β5i ratio: 17; Table 1, Figure 3) and displays broad anti-inflammatory activity [134]. Due to its superior activity in rheumatic diseases and its favorable physicochemical properties compared with ONX 0914 [135,136], it rapidly entered phase I trials for safety evaluation. Although the compound was found to be well tolerated, prednisone and antihistamines were required to prevent systemic drug reactions upon first administration of KZR-616 [137]. In agreement with preclinical investigations that ruled out inhibition of intracellular proteases other than the CP [87], no cytotoxic effects were observed [137]. Furthermore, the immune system was not generally compromised. At an applied dose of 45 mg, KZR-616 blocks ~95% of β5i and ~70% of β2i [137]. This positive rating allowed KZR-616 to enter a phase Ib/II trial in patients suffering from systemic lupus erythematosus with and without lupus nephritis (NCT03393013), and in two phase II studies for active polymyositis or dermatomyositis (NCT04033926), as well as active autoimmune hemolytic anemia or immune thrombocytopenia (NCT04039477). Due to the COVID-19 pandemic, the latter trial was recently withdrawn before first patients were enrolled. Combined, KZR-616 has the potential to advance to a novel and effective therapy for chronic inflammatory diseases in the future and to bring selective iCP inhibition to clinical application. This encouraging development will certainly promote other pharmaceutical companies to raise their budget for proteasome inhibitor research.

## 6. Conclusions

During the last decade, the iCP emerged as a drug target for various diseases. While chronic inflammations and hematological cancers range at the forefront of this development, other potential applications such as treatment of solid tumors, neurodegenerative disorders and infectious diseases either need further exploration or are just entering the stage. Only time can tell us whether the iCP will be approved as a druggable target and whether another chapter can be added to the success story of CP inhibition.

Either way, it is almost certain that the clinical utility of CP inhibitors in general will be expanded in the future, at least in the form of combination therapies to overcome resistances or for the treatment of non-responders. Aside from CP inhibitors, numerous compounds targeting CP-associated proteins, such as the 19S-cap, are in development [138] and may find their application as well, thus guaranteeing the proteasome, in any case, a prospering future.

## Figures and Tables

**Figure 1 cells-10-01929-f001:**
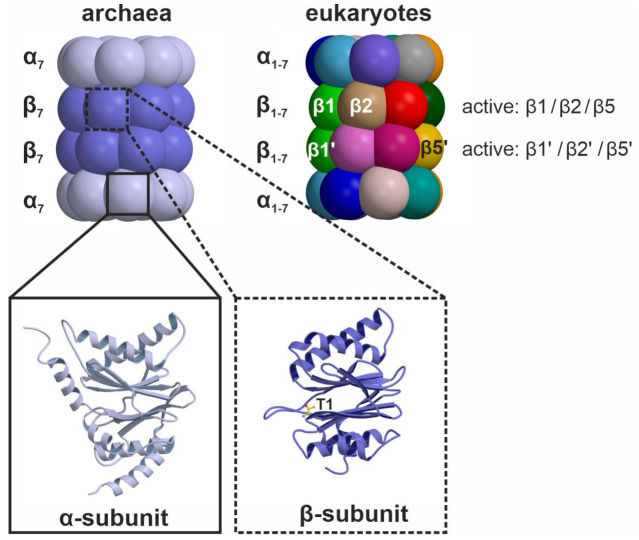
Schematic illustration of archaeal and eukaryotic 20S proteasomes. Simple archaeal 20S proteasome core particles are built of four homo-oligomeric α- and β-rings. Being identical and harboring a catalytic Thr1 residue, all β-subunits are proteolytically active. In contrast, eukaryotic CPs bear hetero-oligomeric α- and β-rings and only three of the seven distinct β-subunits per ring, namely β1, β2 and β5, feature an active site. Note that one β2- and one β5-subunit are hidden in the back of the particle.

**Figure 2 cells-10-01929-f002:**
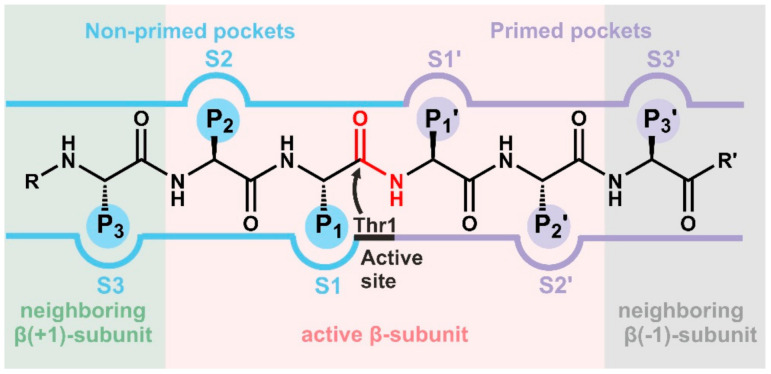
Schematic illustration of the proteasomal substrate binding channel with non-primed (S) and primed (S’) specificity pockets interacting with the amino acid side chains (P-sites) of a peptide. The proteolytically active β-subunit features the active site Thr1 and all other catalytic residues, while the neighboring β-subunits contribute specificity pockets only to the substrate binding channel and do not necessarily feature their own active site.

**Figure 3 cells-10-01929-f003:**
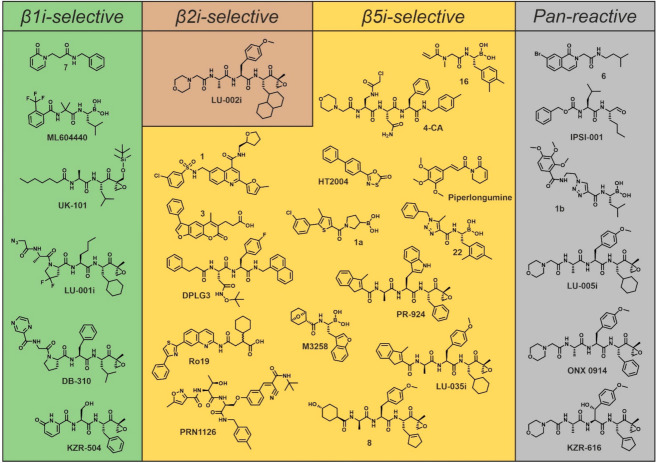
Chemical structures of non-covalently and covalently acting iCP inhibitors. Compounds are grouped and colored according to their subunit selectivity and arranged according to their discussion in the text. The original compound numbering from patents or publications was taken over wherever possible (for references see Table 1). To avoid duplicates, compounds 1a and 1b were given a letter in addition. For inhibition values see Table 1. Due to space limitations, only selected compounds of each class are depicted.

**Table 1 cells-10-01929-t001:** Overview of iCP selective inhibitors and their potency. Due to space limitations, only key compounds of each class are presented. IC_50_ values (given in [µM]) depend on the chosen setting (purified CP/cell lysate/in cells), enzyme concentration and time of incubation, and thus are hardly comparable. As a measure of subunit selectivity, ratios of IC_50_ values are given. The ratio positively correlates with selectivity for the respective i-subunit. ‘n.r.’ designates values that have not been reported. Colors reflect the distinct selectivity patterns of inhibitors: green for β1i-, brown for β2i-, yellow for β5i- and gray for pan-reactive inhibitors. Inhibition values are light-colored, while selectivity ratios are dark-shaded. Most pan-reactive inhibitors target β1i and β5i (green and yellow shades, respectively). Only LU-005i moderately inhibits all immunoproteasome subunits (green, brown and yellow shades).

-	Compound	Source	β1i	β1c	β1c/β1i	β2i	β2c	β2c/β2i	β5i	β5c	β5c/β5i	Reference
β1i-selective	7	Academia	0.021 ^2^	n.r.	n.r.	n.r.	n.r.	n.r.	n.r.	n.r.	n.r.	[71]
	ML604440	Mill. Pharm.	~0.0125	n.r.	n.r.	n.r.	n.r.	n.r.	>1	>1	1	[72]
	UK-101	Academia	0.104	15	144	17	25	1.4	3.1	1	0.3	[73,74] ^1^
	LU-001i	Academia	0.095	24	252	>100	>100	1	20	20	1	[73]
	DB-310	Academia	0.070	0.590	8	n.r.	n.r.	n.r.	>10	>10	1	[75]
	KZR-504	Kezar L. Sci.	0.051	46.35	908	>250	>250	1	4.3	6.9	1.6	[76]
β2i- selective												
	LU-002i	Academia	>100	>100	1	0.18	12.1	67	>100	>100	1	[77]

β5i-selective	1	Academia	n.r.	n.r.	n.r.	n.r.	n.r.	n.r.	1.7	>5	>2	[78]
	3	Academia	>100	>100	1	~100	<100	n.r.	1.6 ^2^	172.2 ^2^	107	[79]
	DPLG3	Academia	>33.3	>33.3	1	>33.3	>33.3	1	0.0045	32.4	7200	[80]
	Ro19	Roche	20	20	1	20 ^3^	n.r.	n.r.	0.025	20	800	[81]
	PRN1126	Princ. Bioph.	>10	>10	1	>10	>10	1	0.0072	0.21	29	[69]
	16	Merck	n.r.	n.r.	n.r.	n.r.	n.r.	n.r.	< 0.05	>5	≥150	[82]
	4-CA	Academia	>100	>100	1	>100	>100	1	0.64	>100	>156	[83]
	HT2004	Academia	n.r.	n.r.	n.r.	n.r.	n.r.	n.r.	1093 ^4^	0.23 ^4^	4752 ^4,7^	[48]
	Piperlongumine	Academia	n.r.	>50	n.r.	n.r.	n.r.	n.r.	15	>50	>3.3	[84]
	1a	Academia	n.r.	n.r.	n.r.	n.r.	n.r.	n.r.	34	102	3	[85]
	22	Genentech	8.5	>20	>2	n.r.	n.r.	n.r.	0.0041	9.1	2219	[65]
	M3258	Merck	>30	>30	1	>30	>30	1	0.0041	2.519	614	[66]
	PR-924	Proteolix	8.2	>30	>3	>30	>30	1	0.022	2.9	131	[86]
	LU-035i	Academia	>10	>10	1	>10	>10	1	0.011	5.5	500	[73]
	8	Kezar L. Sci.	1.85	>25	>13	>25	>25	1	0.034	2.67	78	[87]
Pan	6	Academia	4.9 ^2^	n.r.	n.r.	>100	n.r.	n.r.	4.4 ^2^	n.r.	n.r.	[88]
	IPSI-001	Academia	1.45 ^2^	239 ^2^	164	n.r.	n.r.	n.r.	1.03 ^2^	105 ^2^	101	[89]
	1b	Roche	0.004	0.32	80	20 ^3^	n.r.	n.r.	0.002	0.039	19	[90]
	LU-005i	Academia	0.052	>1	>19	0.47	3.1	6	0.16	3	18	[73,91] ^5^
	ONX 0914	Onyx Phar.	0.46	>10	>21	0.59	1.1	1.8	0.0057	0.054	9	[63,73] ^6^
	KZR-616	Kezar L. Sci.	0.131	>10.6	>80	0.623	0.604	1	0.039	0.688	17	[87]

^1^IC_50_ values were derived from the first reference; the second reference reports the inhibitor. ^2^ K_i_ values instead of IC_50_ values. ^3^ No differentiation between β2c and β2i was made. ^4^ K_inact_/K_i_ values instead of IC_50_ values. In contrast to IC_50_ values, high K_inact_/K_i_ values correspond to high potency. ^5^ The first reference reports the inhibitor, but inhibition values were derived from the second reference. ^6^ The first reference reports the inhibitor but no numerical IC_50_ values for all active sites. Therefore, IC_50_ values were derived from the second reference. ^7^ The 4752-fold selectivity for β5i corresponds to the ratio of β5i/β5c K_inact_/K_i_ values.

## Data Availability

Not applicable.

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
