# Peer review of "A Nut for Every Bolt: Subunit-Selective Inhibitors of the Immunoproteasome and Their Therapeutic Potential"

_cells, 2021, doi:10.3390/cells10081929_

Round 1
Reviewer 1 Report
Huber and Groll provide an excellent overview on the development of subunit-specific inhibitors with a focus on compounds that specifically target subunits of the immunoproteasome. They tackle the topic in an excellent manner from both the basic chemical point of view and demonstrate current data for the translational perspective. The literature review is very intense, the main findings are presented in a comprised logistical manner and the in-depth requirement for specific aspect on the immunoproteasome inhibitor field is reflected in an adequate manner. I recommend publication with a minor revision point stated below:
-lines 68-73: the chronology of the antigen presentation pathway presented by the authors should be revised, e.g. killing mentioned prior to activation of immune defense, rather superficial information presented, which is fine for the purpose of this review, but this also implies the need for correct terminology, “screening” might also be misleading
Author Response
Many thanks for this comment. We have rephrased the respective sentences and also replaced the word "screening".
Reviewer 2 Report
This review is written and organised well. It provides an overview on what is a proteasome, principles of proteasome inhibition, and what is the importance of having inhibitors which recognise the core units of the proteasome. With cumulative research studies on selective as well as the recent pan-immunoproteasome reactive inhibitors, the review provides and extensive coverage of what has been found and published to date.
Some suggestions are as follow:
- The legend for Table 1 can be improved with greater clarity. The columns that are shaded in green/orange/yellow for the Pan inhibitors, do the colours correspond to the respective inhibitors? This is not clearly stated or explained. Additionally, different shades for a single colour was used in the table e.g. light green and dark green, why different shades were used was not clearly stated. Besides, the authors also did not explain why certain values in the table were highlighted while others were not.
- The use of the word 'chapter' in this review should be changed to 'section' instead since this is a review article and not a book volume. e.g. Page 4, line 156 and 160; Page 5, line 182 and 202
Author Response
- We appreciate the comment of the referee and revised the legend of table 1 accordingly. The colors and the different shades are now explained.
- In addition, we replaced the word 'chapter' by 'section' as suggested.